# Automatic Evaluate Dialogue Appropriateness by Using Dialogue Act

**Bao Chen[1]    Yuanjie Wang[1]    Zeming Liu [2]    Yuhang Guo[1*]**

[1]School of Computer Science and Technology, Beijing Institute of Technology, Beijing, China
[2]School of Computer Science and Engineering, Beihang University, Beijing, 100191, China
{chenbao,guoyuhang}@bit.edu.cn    njwangyuanjie@163.com    zmliu@buaa.edu.cn

## Abstract

Evaluation of dialogue systems requires assessing various aspects, among which appropriateness holds significance as a core element of communicative language competence. However, current evaluations heavily rely on human judgments, which are time-consuming, labor-intensive, prone to biases, and lacking objectivity. In this paper, we introduce Dialogue Act Appropriateness (DAA), a novel method that utilizes the underlying patterns of dialogue act transitions to evaluate the appropriateness of chatbot responses. We learn transition patterns from human-human dialogue corpora, evaluating chatbot appropriateness by measuring the similarity of their transition patterns to those observed in human-human dialogues. To validate DAA, we annotate a test dataset by manually evaluating the appropriateness of dialogues from multiple chatbot systems. The experimental results demonstrate a strong correlation between our evaluation metric and human ratings, establishing the reliability of DAA as a measure of dialogue appropriateness.[1]

## 1 Introduction

Automatic evaluation of text generation quality has long been a challenge in the field of natural language processing (NLP) (Liu et al., 2016). When it comes to assessing the quality of dialogue systems, the task becomes even more complex and multifaceted. Dialogue quality encompasses various aspects, including coherence, fluency, engagement, and appropriateness of responses (Mesgar et al., 2020; Mehri and Eskenazi, 2020b; Jiang et al., 2023). Depending on the application and user requirements, different aspects of dialogue quality may need to be evaluated (See et al., 2019). Consequently, there is a growing demand for diverse evaluation metrics capable of capturing the different facets of dialogue quality.

---

[*]Corresponding author.

[1]The code and data is available at https://github.com/xba0/DAA

Appropriateness is a fundamental characteristic of language usage, serving as a core element of communicative language competence and an evaluative criterion for language users or learners (Grundy, 2008). It encompasses the notion of using language in a manner that is suitable, contextually relevant, and socially acceptable in various communication settings. Evaluating the appropriateness of language use is crucial for assessing the proficiency and effectiveness of individuals' communicative skills in both spoken and written forms of expression. However, the current assessment of appropriateness mainly relies on human judgment, which is not only time-consuming and laborious, but also leads to potential bias and lack of objectivity.

According to Fetzer (2004), the concept of appropriateness is closely tied to the effective execution of dialogue acts. Appropriateness serves as a necessary and sufficient condition for carrying out dialogue acts properly. These acts play a crucial role in identifying the actions performed by utterances in a conversation, shedding light on their intended meaning and purpose in terms of illocutionary force, as discussed by Searle (1970). By analyzing dialogue acts, researchers can gain a deeper understanding of the interactions between speakers and the objectives they aim to accomplish. This comprehension is further enhanced through the works of Allwood et al. (1992); Stolcke et al. (2000), who emphasize the significance of dialogue acts in unraveling the functions, intended meanings, and purposes of utterances. Inspired by this understanding, we consider the dialogues in human-human corpora to exhibit a high level of appropriateness, and the dialogue act pattern within these conversations is deemed suitable. Therefore, we can leverage human-human dialogues as a guiding reference to assess the appropriateness of human-chatbot dialogues, considering the patterns of dialogue act transitions.

Specifically, we propose an automatic evaluation method called Dialogue Act Appropriateness (DAA) to assess appropriateness based on dialogue acts. Initially, we employ a dialogue act classifier to categorize dialogue utterances into sequences of dialogue acts. Subsequently, we model the dialogue act transition patterns in human-human dialogue corpora to capture the typical patterns of dialogue act transitions, which serve as the reference dialogue act patterns. Finally, we compare the dialogue act transition patterns of the chatbot's responses with the reference dialogue act patterns to measure the appropriateness of the chatbot's responses.

To the best of our knowledge, there is currently a lack of annotated dialogue datasets targeting pragmatic appropriateness. To facilitate the automated evaluation of pragmatic appropriateness and validate the efficacy of DAA, we curated an evaluation dataset. The dataset comprises conversation logs from six evaluation objects, including five popular chatbot systems and one human for comparison. Following the data collection process, seven annotators independently assigned ratings to evaluate the appropriateness of each turn of responses from the objects.

Experimental results demonstrate a strong correlation between the appropriateness scores generated by DAA and the annotators. This indicates that DAA effectively captures the notion of appropriateness in dialogues and aligns well with human perceptions.

Our contributions are summarized as follows:

- We quantify human subjective evaluation of dialog appropriateness using an **interpretable** approach, providing a **stable** evaluation metric called Dialogue Act Appropriateness (DAA).

- To evaluate the effectiveness of DAA and facilitate further research, we conduct human evaluations on six different evaluation objects and build **a dedicated test dataset** for evaluating **pragmatic appropriateness**.

- Experiments show a **strong correlation** between our proposed metric and human judgments, demonstrating the effectiveness of DAA in accurately assessing dialog appropriateness.

## 2   Method

We posit that human-human dialogues exhibit a higher level of appropriateness in the use of dialogue acts. Therefore, by measuring the similarity between the dialogue act patterns observed in human-chatbot dialogues and those in human-human dialogues, we can assess the appropriateness of chatbot responses. The process of DAA, as depicted in Figure 1, involves three key steps. Firstly, we classify the utterance sequences in a multi-turn dialogue into sequences of dialogue acts, enabling us to learn and compare the transition patterns of dialogue acts. Next, we model the transition patterns of dialogue acts among interlocutors using a human-human dialogue corpus. This allows us to capture the inherent patterns of dialogue act transitions that occur in human conversations, serving as a reference for assessing appropriateness. Finally, we compare the dialogue act transition patterns observed in a given human-chatbot dialogue with the learned patterns from the previous step, providing an appropriateness score based on the degree of similarity.

### 2.1   Dialogue Act Classification

We trained a dialogue act classifier to map the utterance sequence in a dialogue to a sequence of dialogue act labels, allowing us to model the dependency among dialogue acts. A context-aware dialogue act classifier enhances the accuracy of dialogue act classification, partially due to the classifier's utilization of dialogue act transition patterns present in DA training data. Our approach involves evaluating the similarity of these transition features across distinct corpora, such as human-human dialogues and human-chatbot dialogues, to establish scoring criteria. So, we deliberately restrict the DA classifier from utilizing contextual information to prevent potential biases arising from the classifier learning and incorporating prior transition patterns from the DA dataset. Specifically, we fine-tuned a transformer-based pre-trained model as our classifier. We concatenated the [CLS] token with the utterance in the dialogue as the input to the pre-trained model, considering the final hidden state of the [CLS] token as the representation of the entire sequence. The representation $h_{cls}$ of the [CLS] token underwent linear projection and tanh activation, followed by a softmax classifier to obtain the probabilities of dialogue act labels. We used cross-entropy as the loss function in the training

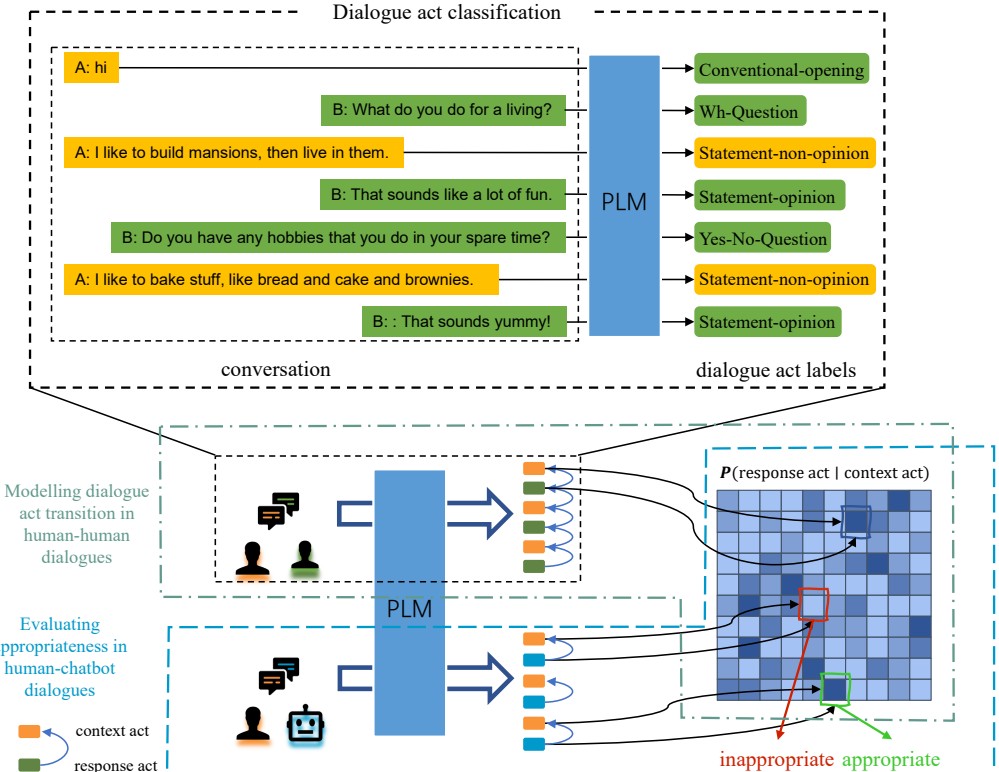

Figure 1: An illustration of DAA[2]. The colored matrix represents the identified patterns of dialogue act transitions from human-human conversation corpora. Darker cells indicate a more frequent occurrence of these dialogue act transitions in human dialogues.

process.

## 2.2 Modelling Dialogue Act Transition in Human-Human Dialogues

In this step, we utilize a well-trained dialogue act classifier to annotate dialogue act labels from human-human dialogues and calculate the conditional probability distribution of response acts given the dialogue history. A dialogue can be divided into n turns, where two speakers take turns speaking. Odd turns and even turns are generated by different individuals. Each turn consists of multiple utterances spoken by the same speaker. We denote the j-th utterance in the i-th turn of the dialogue as $utt_{i,j}$. Following the method described in section 2.1, we assign a dialogue act label (DA) to $utt_{i,j}$.

Considering that an utterance often responds to the last utterance from the other speaker, we overlook the weak connections between dialogue acts in non-adjacent turns. Our focus is on assessing the appropriateness of response acts relative to the dialogue act of the dialogue partner. The first ut-

terance of a turn usually serves as a continuation of the previous speaker's content, while the subsequent utterances typically complement the first utterance. Therefore, we do not consider the transition relationships of dialogue acts within the same turn.

Hence, we treat the last utterance of the speaker in a turn, $utt_{i,-1}$, as the context, and the first utterance of the next turn, $utt_{i+1,1}$, as the response to this context. From a dialogue consisting of $N$ turns, we obtain $N-1$ context-response pairs. We then calculate the conditional probability $P(DA(response) \mid DA(context))$ from human dialogues. Finally, we establish a transition probability matrix for dialogue acts at the turn boundary.

## 2.3 Evaluating Appropriateness in Human-Chatbot Dialogues

To measure the appropriateness of a chatbot's responses, we first use the same dialogue act classifier as in section 2.2 to obtain the sequence of dialogue acts in the dialogue. To focus on evaluating the chatbot's responses in the evaluation process, we mask out the human responses to the chatbot and only consider the chatbot's responses to humans.

---

[2]A detailed list of dialogue act labels can be found at http://compprag.christopherpotts.net/swda.html#tags

We treat the last utterance of a human turn as the context and the first utterance of the subsequent bot turn as the response to this context. We then use the conditional probability of dialogue act transition to measure the appropriateness of each context-response pair:

$$DAA_i = P(DA(response_i)|DA(context_i))$$

Finally, the appropriateness at the dialogue level is computed by taking the geometric mean of the appropriateness scores at the turn level, where $N$ represents the number of context-response pairs:

$$DAA = \sqrt[N]{\prod_{i=1}^{N} DAA_i}$$

## 3 Evaluation Data Construction

While annotated data exists for overall appropriateness and semantic appropriateness, to the best of our knowledge, we did not come across any annotated data specifically designed for evaluating pragmatic appropriateness in conversations. In order to assess the effectiveness of DAA, we gathered conversation logs from six objects with different levels of intelligence. Each response of these conversations is manually annotated with a score indicating its appropriateness.

### 3.1 Evaluation Objects

In our experiment, we selected five famous chatbots as the evaluation objects: *blender* (Roller et al., 2021), *cleverbot*[3], *jabberwacky*[4], *meena* (Adiwardana et al., 2020) and *mitsuku*. Additionally, we included a human for comparison. We collected 50 dialogues from each object. All dialogues were sourced from publicly available data. The dialogues of *Meena*, *Mitsuku*, and *Human* were obtained from Google Research[5], while the dialogues of *Blender* were sourced from the ParlAI platform[6].

### 3.2 Manual Annotation

Following Mehri and Eskenazi (2020a), we employed a 5-point Likert scale for the manual quality assessment of appropriateness. In this process, we provided annotators with a chatbot response and the preceding utterance of the conversational partner as the context. Their responsibility was to assess the appropriateness of the provided chatbot response, assigning an integer score within the range of 0 to 4, where 4 represents "very appropriate" and 0 signifies "very inappropriate"

We gathered 300 dialogues involving six entities, resulting in a total of 2,716 responses. We employed seven annotators who individually assigned turn-level scores to all the collected responses. The average score for each turn within a dialogue was used to as the dialogue-level score, and the average score of all dialogues pertaining to an object was used as the object's score.

To examine the inter-annotator consistency, we calculated the Pearson and Spearman correlation coefficients between the scores of one annotator and the mean scores of other annotators for different chatbots. The average Pearson correlation coefficient was found to be 96.19%, and the average Spearman correlation coefficient was 92.65%.

## 4 Experimental Setup

### 4.1 Training Datasets

The dialogue act classifier utilized in our study is trained on the Switchboard dialogue act corpus (SWDA) (Stolcke et al., 2000). The transition probability matrices for dialogue act transfer were computed based on different human-human dialogue datasets, namely DailyDialog (Li et al., 2017), Personachat (Zhang et al., 2018), Reddit Corpus (Lee et al., 2019), and Cornell Movie-Dialogs Corpus (Danescu-Niculescu-Mizil and Lee, 2011). These diverse datasets are chosen to explore the impact of dialogue domains and styles.

### 4.1.1 Dataset for Dialogue Act Classification

The switchboard dialogue act corpus (Stolcke et al., 2000) is a widely used dataset for studying dialogue act classification. It consists of telephone conversations recorded between two participants, covering a wide range of topics and conversational styles. Each utterance in the corpus is annotated with a dialogue act label, indicating the intention or function of the speaker's utterance. We chose the SWDA as our training data due to its content's close resemblance to daily conversation. To facilitate our experiments, we utilized a preprocessed version of the dataset, which can be found here[7]. After preprocessing, the dataset consisted of a total

---

[3] https://www.cleverbot.com/j2conversations
[4] http://www.jabberwacky.com/
[5] https://github.com/google-research/google-research/tree/master/meena
[6] https://parl.ai/projects/recipes/

[7] https://github.com/NathanDuran/Switchboard-Corpus

of 199,740 utterances. The maximum length of an utterance in the dataset was 132, while the mean length was 9.62.

### 4.1.2 Human-Human Dialogue Datasets

**Daily Dialog** (Li et al., 2017) is a high-quality multi-turn dialogue dataset that contains conversations about daily life. It is cleaner compared to other corpora and covers a wide range of topics for casual conversation. The dataset includes a total of 13,118 dialogues, with an average of 7.9 speaker turns per dialogue. The average number of tokens per dialogue is 114.7, and the average number of tokens per utterance is 14.6.

**Personachat** (Zhang et al., 2018) is a crowd-sourced dataset collected through Amazon Mechanical Turk. In this dataset, each pair of speakers conditions their dialogue on a given profile. The dataset consists of 1,097 dialogues with a total of 162,064 utterances.

**Reddit Corpus** (Lee et al., 2019) is a dataset used in Task 2 of the DSTC 8 Competition. The data is crawled from 1,000 relatively non-toxic sub-reddits on Reddit, covering a period of 12 months from November 2017 to October 2018. The training set contains 5,085,113 dialogues.

**Cornell Movie-Dialogs Corpus** (Danescu-Niculescu-Mizil and Lee, 2011) is a large metadata-rich collection of fictional conversations extracted from raw movie scripts. It includes 220,579 conversational exchanges and a total of 304,713 utterances.

### 4.2 Implementation Details

We employed the base and large versions of BERT (Devlin et al., 2019), RoBERTa (Liu et al., 2019), BART (Lewis et al., 2020), and Ernie 2.0 (Sun et al., 2020) as the dialogue act classifiers. This choice was made based on several factors, including their popularity in the field, their strong performance in various natural language processing tasks, and their availability as pre-trained models. BART, BERT, and RoBERTa were obtained from Hugging-face Transformers[8], while Ernie 2.0 was obtained from PaddleHub[9]. These transformer-based models were fine-tuned for 10 epochs, and the classifiers that demonstrated superior performance on the validation set were selected for subsequent experiments. Given that the longest sentence in the SWDA dataset consists of 132 words, we set the

maximum input sequence length to 256 to accommodate the data. To extract appropriateness features in human dialogues, we utilized the NLTK toolkit [10] to split each dialogue turn into utterances and annotated these utterances using dialogue act classifiers.

## 5 Results and Analysis

In this section, we present the correlation of DAA with human ratings and conduct a comparative analysis with other appropriateness metrics, then address the following research questions (RQs): (1) Does DAA provide a stable and objective evaluation of dialogue appropriateness? (2) Is DAA sensitive to the accuracy of the dialogue act classifier? (3) How does the domain of the training data impact the evaluation of dialogues? (4) How can we interpret the appropriateness scores provided by DAA? (5) How does the direction of context-response pairs affect the performance of DAA?

### 5.1 Correlation with Human Ratings

Following previous work (Jiang et al., 2022), we employed Pearson and Spearman correlation coefficients to assess the performance of the proposed method. The Pearson coefficient is used to measure the linear correlation between two continuous variables, while the Spearman coefficient evaluates the statistical dependence between the rankings of two variables. Higher correlation coefficients indicate a stronger alignment between DAA and human ratings. Table 1 and Table 2 present the Pearson and Spearman correlation coefficients between DAA and human ratings. We conducted extensive experiments using multiple pre-trained models and corpora. The results demonstrate a strong correlation between DAA and human evaluation, confirming the effectiveness of DAA. The performance of the base model is weaker compared to the large model, despite their similar classification accuracy. This difference can be attributed to the large model's ability to capture more comprehensive features, resulting in improved performance. The best performance was achieved by employing the RoBERTa-large model as the dialogue act classifier and using the PersonaChat corpora.

---

[8] https://github.com/huggingface/transformers
[9] https://github.com/PaddlePaddle/PaddleHub

[10] http://www.nltk.org/

| Model | Daily Dialog | Persona Chat | Reddit | Cornell |
|---|---|---|---|---|
| $Bart_b$ | 85.05 | 83.34 | 79.69 | 76.39 |
| $Bert_b$ | 93.28 | 92.62 | 88.59 | 82.29 |
| $Ernie_b$ | 90.95 | 79.38 | 84.40 | 79.59 |
| $RoBERTa_b$ | 91.45 | 86.43 | 85.49 | 88.15 |
| $Bart_l$ | 92.54 | 93.22 | 88.67 | 83.15 |
| $Bert_l$ | 92.83 | 92.54 | 91.28 | 86.58 |
| $Ernie_l$ | 92.15 | 88.63 | 82.75 | 74.65 |
| $RoBERTa_l$ | 97.66 | **98.70** | 93.60 | 92.17 |

Table 1: Pearson correlation coefficients (%) between human ratings and DAA. All results are statistically significant with p-value < 0.05

| Model | Daily Dialog | Persona Chat | Reddit | Cornell |
|---|---|---|---|---|
| $Bart_b$ | 77.14 | 88.57 | 77.14 | 71.42 |
| $Bert_b$ | 88.57 | **94.28** | 88.57 | 77.14 |
| $Ernie_b$ | **94.28** | 82.85 | 88.57 | 88.57 |
| $RoBERTa_b$ | **94.28** | **94.28** | 88.57 | 88.15 |
| $Bart_l$ | 88.57 | 88.57 | 88.57 | 88.57 |
| $Bert_l$ | 88.57 | 88.57 | 88.57 | 88.57 |
| $Ernie_l$ | **94.28** | **94.28** | **94.28** | 71.42 |
| $RoBERTa_l$ | **94.28** | **94.28** | 88.57 | 88.57 |

Table 2: Spearman correlation coefficients (%) between human ratings and DAA. All results are statistically significant with p-value < 0.05

## 5.2 Comparison with Other Appropriateness Metrics

In Table 3, we compared DAA with a popular method FED (Mehri and Eskenazi, 2020a) and the recently proposed C-PMI method (Ren et al., 2023) on our dataset. Since these methods do not evaluate the aspect of pragmatic appropriateness, we conducted a comparison using their semantic appropriateness dimension. Our proposed DAA approach achieved the highest correlation with human ratings across turn-level, dialogue-level, and system-level evaluations. It is worth noting that C-PMI has made improvements over the FED method, enhancing its performance in terms of semantic appropriateness (as reported in their paper). However, in the dimension of pragmatic appropriateness, which is our focus, it exhibited a performance decline.

## 5.3 RQ 1: Stability

Table 4 presents the Pearson correlation coefficients between the ratings of annotator $a_i$ and the mean ratings of other annotators. We compare them with DAA ratings obtained from Roberta-large on the PersonaChat dataset. It can be observed that the

| Method | Turn-Level | | Dialogue-Level | | System-Level | |
|---|---|---|---|---|---|---|
| | $r$ | $\rho$ | $r$ | $\rho$ | $r$ | $\rho$ |
| FED | 14.06 | 12.99 | 16.50 | 11.81 | 55.29 | 42.86 |
| C-PMI | 4.51 | 4.72 | 2.83 | 2.61 | 13.21 | 8.57 |
| DAA | **21.67** | **21.19** | **36.16** | **36.88** | **89.05** | **87.38** |

Table 3: A comparison with the popular method FED (Mehri and Eskenazi, 2020a) and the recently proposed C-PMI method (Ren et al., 2023) on our dataset. Here, $r$ and $\rho$ represent the Pearson correlation coefficient and Spearman correlation coefficient, respectively.

| | DAA | $a_i$ |
|---|---|---|
| $A-a_1$ | 97.73 | 96.90 |
| $A-a_2$ | 97.85 | 95.23 |
| $A-a_3$ | 97.13 | 89.26 |
| $A-a_4$ | 96.68 | 97.81 |
| $A-a_5$ | 97.41 | 96.73 |
| $A-a_6$ | 97.33 | 99.40 |
| $A-a_7$ | 97.85 | 98.02 |
| Average | **97.43** | 96.19 |

Table 4: The Pearson correlation coefficients (%) between the ratings of annotator $a_i$ and the mean ratings of other annotators. DAA is included for comparison.

DAA method exhibits a relatively stable correlation with the mean ratings of other annotators, ranging from 96.68% to 97.85%. In contrast, the human annotators demonstrate more significant fluctuations in their correlations with the mean ratings, ranging from 89.26% to 99.40%. These findings indicate that DAA can provide a stable and objective evaluation of dialogue appropriateness.

## 5.4 RQ 2: The Effect of the Dialogue Act Classification Accuracy

We experimented with eight pre-trained models as dialogue act classifiers. The models demonstrate relatively high accuracy on both the validation set and test set when considering only a single utterance as input, without incorporating additional context features. Our focus lies in capturing the transition patterns of dialogue behaviors rather than specific dialogue act labels. With an accuracy of at least 73%, all models are capable of annotating dialogue acts. To investigate the impact of classifier accuracy on DAA, we analyze the relationship between classifier accuracy and the correlation of DAA with human ratings. The Pearson correlation between DAA and different levels of classifier accuracy is illustrated in Figure 2. A linear fit was attempted, yielding an R-squared value of 0.063. This indicates that the effectiveness of DAA is not

significantly influenced by the accuracy of the dialogue act classifier.

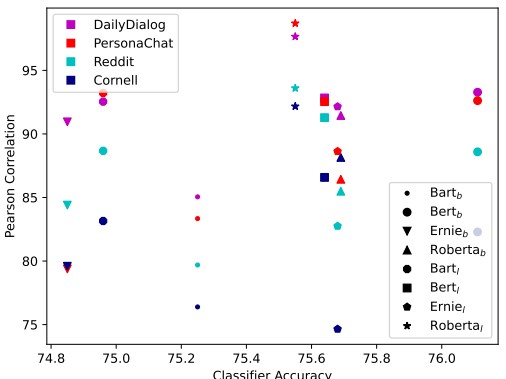

Figure 2: The scatterplot illustrates the Pearson correlation of DAA with different test accuracy of dialogue act classifiers.

## 5.5 RQ 3: The Impact of Corpus Domain

It can be observed from Table 1 and 2 that using DailyDialog and PersonaChat as training datasets for the dialogue act binary model yields stronger correlations compared to the Reddit Corpus and Cornell Movie-Dialogs Corpus. The dialogue data in DailyDialog and PersonaChat is more closely aligned with chitchat, in contrast to the other two datasets. This indicates that the domain of the training data has an impact on the evaluation. We can assess specific dialogues more effectively by employing corresponding corpus that belong to the same domain. For instance, if we intend to evaluate the quality of a healthcare chatbot, we can utilize a healthcare dialogue corpus as the training data.

## 5.6 RQ 4: Case Study

In Table 5, we present a snippet of a conversation log selected from the publicly available human-chatbot dialogue logs from (Adiwardana et al., 2020). Due to the length of the original conversation, we have extracted a portion for illustration purposes. For each first response from the chatbot in a turn, we provide both manual ratings and DAA automatic ratings to assess its appropriateness. To facilitate comparison, all ratings are scaled within the range of 0 to 1. Overall, there is a close correspondence between the manual ratings and the DAA scores. In the second human turn, the human poses a Yes-No-Question, and the bot responds appropriately, resulting in high ratings from both the human annotator and DAA. However, in the third

human turn, the human poses a Yes-No-Question, but the bot's response veers off-topic by asking an unrelated question, leading to low ratings from both the human annotator and DAA. It is noteworthy that the dialogue act classifier erroneously labeled the dialogue act of the third bot turn as "Wh-Question" instead of the correct label. However, this misclassification does not impact the consistent ratings assigned by DAA, as we employed the same dialogue act classifier for annotating the chatbot's dialogue logs and for modeling appropriateness using the training corpus. Thus, such classification errors are mitigated during the modeling process. Our DAA method allows for pinpointing specific inappropriate response utterances and provides a human-understandable reason (inappropriate dialogue act) for the assessment.

## 5.7 RQ 5: Ablation Study

Human-chatbot dialogues involve two types of turn transitions: from the bot to the human and from the human to the bot. As mentioned in Section 2.3, to focus on assessing bot responses, DAA evaluates only the bot's reply to human. In Table 6, we compare the effect of considering the direction of responses on the performance of DAA. We measure the effectiveness and stability of the models by calculating the mean and standard deviation of the Pearson and Spearman correlation coefficients between DAA's scores and human ratings. We evaluate the performance of both approaches on different corpora. The results demonstrate that considering the direction of responses yields the best performance, maintaining the highest correlation with human ratings and exhibiting greater stability. Notably, DAA shows minimal performance loss when trained on the Reddit and Cornell corpora, both of which encompass domains distinct from the evaluation objects.

## 6 Related Work

Early dialogue evaluation methods employed statistical metrics such as BLEU, METEOR, and ROUGE (Papineni et al., 2002; Banerjee and Lavie, 2005; Lin, 2004). These metrics measure word overlap between system-generated and reference responses. Embedding-based methods like BERTScore (Zhang et al., 2020a) utilize contextual embeddings to compute the similarity between system responses and references. ADEM (Lowe et al., 2017) collects human-scored data and employs a hi-

| Role | Sentence | Dialogue act label | Human | DAA |
|---|---|---|---|---|
| **Human:** | hi | Conventional-opening | | |
| **Bot:** | Hi there. | Conventional-opening | 0.87 | 0.85 |
| **Human:** | do you have plan for thanksgiving weekend? | Yes-No-Question | | |
| **Bot:** | No I don't have it. | Statement-non-opinion | 1.00 | 1.00 |
| | But I do have an answer to every question.. | Statement-non-opinion | | |
| **Human:** | do you have suggestion for the thanksgiving holidays? | Yes-No-Question | | |
| **Bot:** | Where would I get it? | Signal-non-understanding | 0.04 | 0.01 |
| | But I do have an answer to every question.. | Statement-non-opinion | | |
| **Human:** | do you know thanksgiving? | Yes-No-Question | | |
| **Bot:** | I know very little about it. | Statement-non-opinion | 0.96 | 1.00 |
| | What kind of it do you like? | Wh-Question | | |
| **Human:** | do you like turkey? | Yes-No-Question | | |
| **Bot:** | Yes but it can be a bit dry. | Statement-non-opinion | 0.96 | 1.00 |
| | I only really eat it at Christmas. | Statement-non-opinion | | |

Table 5: An example of human-chatbot dialogue intercepted from our dataset. The appropriateness of each first utterance from the bot turn is rated by the human and our DAA method, scaled from 0 to 1 for comparison.

| | DailyDialog | | PersonaChat | | Reddit | | Cornell | |
|---|---|---|---|---|---|---|---|---|
| Method | $r$ | $\rho$ | $r$ | $\rho$ | $r$ | $\rho$ | $r$ | $\rho$ |
| DAA | 91.99±3.25 | 90.00±5.53 | 89.36±5.79 | 90.71±3.98 | 86.81±4.29 | 87.85±4.46 | 82.87±5.58 | 82.80±7.52 |
| w/o dir. | 82.85±2.85 | 82.85±5.71 | 84.93±6.00 | 82.85±11.43 | 63.36±9.61 | 38.57±21.71 | 67.63±6.98 | 47.85±13.83 |

Table 6: The mean and standard deviation of Pearson correlation coefficient ($r$) and Spearman correlation coefficient ($\rho$) between DAA scores and human ratings when considering or not considering the direction of context-response pairs.

erarchical RNN to predict scores based on dialogue context, reference, and system response. RUBER (Tao et al., 2018) integrates referenced and unreferenced indicators. However, these methods rely on pre-given references or manual scoring, which is impractical in dialogue evaluation due to the diverse range of possible responses. In contrast, DAA does not require manual scoring or pre-provided reference responses for evaluating dialogue quality.

Some researchers have proposed unsupervised metrics to reduce reliance on manual annotations, such as GRADE (Huang et al., 2020), DynaEval (Zhang et al., 2021), and QuantiDCE (Ye et al., 2021). These methods adopt a response-differentiating paradigm, using extensive human-to-human dialogues as positive examples and applying heuristic perturbations to obtain negative examples. The models are trained to distinguish between paired positive and negative examples. However, these heuristic perturbations are often simplistic, such as random utterance order shuffling, limiting the model's ability to learn accurate decision boundaries. On the other hand, PONE (Lan et al., 2020) performs negative example sampling based on semantic similarity with positive examples. Park et al. (2021) use a masked language model to replace some words in the response, considering the dialogue history, with words unre-

lated to the dialogue history to generate negative examples. DEAM (Ghazarian et al., 2022) analyzes coherent error categories in real dialogues and designs targeted automatic rules for constructing negative examples. FineD-Eval (Zhang et al., 2022) supplements training data with additional model scores, such as question-answer relevance scoring and natural language inference contradiction scores, to obtain positive and negative samples. Compared to these methods, DAA directly models human dialogues without the need for negative sample generation.

Some methods directly obtain scores based on the token probability distribution provided by language models. HolisticEval (Pang et al., 2020) uses GPT2 (Brown et al., 2020) to calculate the conditional probability of a response given the dialogue history. Mehri and Eskenazi (2020a) utilize DialoGPT (Zhang et al., 2020b) to pre-given follow-up utterances and take the likelihood as scores. FULL (De Bruyn et al., 2022) improves upon Mehri and Eskenazi (2020a) by validating a large number of follow-up utterances and selecting the follow-up utterance that has the highest correlation with human evaluation. In comparison to these methods, we do not model token-level inputs. Instead, we model dialogue act sequences.

In the field of evaluation, there is variability

in the definition of appropriateness. Lowe et al. (2017) focus on the overall impression of the response. Young et al. (2018) define appropriateness as whether the response is appropriate in terms of grammar, topic, and logic. Liang and Li (2021) consider appropriateness to be synonymous with the overall quality. Mehri and Eskenazi (2020a) emphasize semantic appropriateness. Stasaski and Hearst (2023) associate appropriateness with dialogue act. In this work, we adopt the definition of appropriateness from Stasaski and Hearst (2023) and leverage the dialogue act to measure the appropriateness.

## 7 Conclusion

In this paper, we propose an interpretable and effective method to assess the appropriateness of dialogue agent responses by modeling dialogue act transitions. To facilitate the automated evaluation of pragmatic appropriateness, we annotated a manually scored dataset consisting of 2,716 turn-level appropriateness ratings for six different agents, with evaluations provided by seven annotators. Our extensive experiments, using eight pre-trained language models and four human conversation corpora, demonstrate a strong correlation between our evaluation method and human ratings. These results confirm the reliability and effectiveness of DAA in assessing the appropriateness of dialogue agent responses.

## Limitations

While our method demonstrates a strong correlation with human evaluations, this work has certain limitations. Firstly, like other fine-grained evaluation metrics, our approach focuses on one aspect of a conversation while neglecting other aspects. Comprehensively evaluating all facets of dialogues using a single method is a challenging task. To achieve a more holistic evaluation, future research can explore the integration of semantic-based metrics with our approach. Furthermore, constrained by the existing dialogue act classification datasets, our employed DA labels may not distinguish nuanced intentions. It is important to note that our approach is not restricted to any specific dialogue act classification dataset and can seamlessly transition to finer-grained dialogue act datasets in the future. Lastly, the performance of this method is influenced by the dialogue corpus used. Fortunately, it is possible to migrate to dialogues in different domains at a low cost, by collecting unannotated dialogues from the relevant domain and automatically labeling them using a DA classifier.

## Ethics Statement

In this paper, we propose a novel automatic evaluation method for assessing the appropriateness of dialogues. The study utilized publicly available dialogue corpora and human-chatbot dialogue logs. The data collection process strictly adhered to legal and ethical requirements, ensuring privacy and consent. Participants were provided with comprehensive information about the study's purpose, data collection procedures, and potential risks or benefits. They had the opportunity to ask questions and voluntarily provide their consent before participating. The collected data were used solely for research purposes.

## Acknowledgments

Thanks for the insightful comments from reviewers. This work is supported in part by the National Key R&D Program of China (No. 2020AAA0106600) and the National Natural Science Foundation of China (No.U21B2009)

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
