# OpenReview forum: "Automatic Evaluate Dialogue Appropriateness by Using Dialogue Act"
_EMNLP/2023/Conference — EMNLP 2023 Findings_

### Official Review · Reviewer_gNvL · 2023-08-02

**Soundness:** 3

**Excitement:**

4: Strong: This paper deepens the understanding of some phenomenon or lowers the barriers to an existing research direction.

**Missing References:**

Nothing is missing.

**Paper Topic And Main Contributions:**

The paper concentrates on an important aspect of dialogue systems - appropriateness. A novel method is introduced to evaluate dialogue agent responses by modeling dialogue act transitions - Dialogue Act Appropriateness.  The method is validated, the experimental results demonstrate a strong correlation with manual evaluation.
The study utilizes publicly available dialogue corpora and human-chatbot dialogue logs.
5 tables and 2 figures illustrate the discussion.


**Questions For The Authors:**

No questions.

**Reasons To Accept:**

The strengths:
• Human subjective evaluation of dialogue appropriateness is quantified by using an approach, that provides a stable evaluation metric - Dialogue Act Appropriateness.
 • To evaluate the effectiveness of the proposed method, human evaluations on 6 different dialogue agents are conducted, and a test dataset is built for evaluating dialogue pragmatic appropriateness.
• Experiments using 8 pre-trained language models and 4 human conversation corpora show a strong correlation between the proposed metric and human judgments.


**Reasons To Reject:**

The paper can be accepted.

**Reproducibility:**

3: Could reproduce the results with some difficulty. The settings of parameters are underspecified or subjectively determined; the training/evaluation data are not widely available.

**Reviewer Confidence:**

3: Pretty sure, but there's a chance I missed something. Although I have a good feel for this area in general, I did not carefully check the paper's details, e.g., the math, experimental design, or novelty.

**Typos Grammar Style And Presentation Improvements:**

Style is fine.
Minor spelling errors, e.g.: work. our (line 581)

---

> ### Author Rebuttal · Authors · 2023-08-29
>
> Thanks for the thorough review for acknowledging our proposed framework.
>
> **For Minor spelling errors:**
>
> Thanks for your comments. We have proofread and revised the manuscript
> thoroughly according to your suggestions.

---

### Official Review · Reviewer_LjnF · 2023-08-05

**Soundness:** 2

**Excitement:**

3: Ambivalent: It has merits (e.g., it reports state-of-the-art results, the idea is nice), but there are key weaknesses (e.g., it describes incremental work), and it can significantly benefit from another round of revision. However, I won't object to accepting it if my co-reviewers champion it.

**Paper Topic And Main Contributions:**

This paper proposed a new metric to evaluate dialogue appropriateness via calculating the geometric mean of transition probabilities of dialog act context-response pair. The study is conducted on 2716 turn-level appropriateness ratings across 6 different agents with 7 annotators. Extensive experimental results show that the proposed DAA metric is strongly correlated with human ratings.

**Questions For The Authors:**

1.  Could you explain more about ignoring the direction of context-response?


**Reasons To Accept:**

The proposed evaluation method considers the simple perplexity-like metric for dialogue act sequence as the perplexity for a sentence. Experiments are very extensive.

**Reasons To Reject:**

1. Similar to perplexity, the proposed method will be strongly impacted by the training domain on both the dialogue act classifier and the sequential pattern of the dialogue act. Hence, in real use cases, domain adaptation for dialogue act classifier and a very amount of in-domain human data are needed for the evaluation.
2. it is unknown how the single 5-point Likert scale is annotated. This single subjective human evaluation score is not very convincing for dialogue evaluation. Also, considering a new dimension of human evaluation, without any calibration, I cannot imagine the proposed method will still be highly correlated with human annotation. Hence, this method could only be used as a preliminary evaluation of perplexity.

**Reproducibility:**

2: Would be hard pressed to reproduce the results. The contribution depends on data that are simply not available outside the author's institution or consortium; not enough details are provided.

**Reviewer Confidence:**

3: Pretty sure, but there's a chance I missed something. Although I have a good feel for this area in general, I did not carefully check the paper's details, e.g., the math, experimental design, or novelty.

---

> ### Author Rebuttal · Authors · 2023-08-29
>
> Thanks for the thorough review for acknowledging our proposed framework.
>
> **Reasons To Reject 1: the proposed method will be strongly impacted by the training domain**
>
>
> Our method encompasses the Dialogue Act (DA) classification phase and the DA sequence modeling phase.
>
> In Section 5.3, we illustrate that the accuracy of the classifier has a relatively minor effect on the method's relevance. This is due to the fact that classification errors are uniformly applied to both the modeling corpus and the dialogues under evaluation. It's noteworthy that we leverage the transitions of DAs rather than specific DA labels.
>
> Furthermore, the modeling of DA sequences in human-human dialogues involves utilizing unlabeled dialogues specific to the corresponding domain. These dialogues are automatically labeled using the DA classifier, thereby eliminating the need for additional manual annotations.
>
> As a result, it is feasible and cost-effective to adapt our method to different dialogue domains, requiring only minimal domain-specific unlabeled data.
>
>
>
> **Reasons To Reject 2-1:  it is unknown how the single 5-point Likert scale is annotated.**
>
> Following Mehri and Eskenazi (2020), we employed a 5-point Likert scale for the manual quality assessment of appropriateness. In this process, we presented annotators with both human utterances and bot responses at the turn level. Their task was to evaluate the appropriateness of the given bot response. They assigned an integer score ranging from 0 to 4, where 4 represents "very appropriate" and 0 signifies "very inappropriate."
>
>
>
> **Reasons To Reject 2-2:  This single subjective human evaluation score is not very convincing for dialogue evaluation.**
>
> To mitigate the inherent subjectivity inherent in individual human scores, we adopted an averaging approach. Specifically, we calculated the mean score assigned by annotators to the same response. This strategy served to temper the subjectivity in individual scores.
>
>
> Furthermore, following the methodology of Mehri and Eskenazi (2020) and as outlined in section 3.2 of our paper, we conducted computations of Pearson and Spearman correlations. These correlations, measuring the consistency between annotators' mean scores and the scores assigned by other annotators, yielded values of 0.9619 and 0.9265 respectively. These robust correlations attest to the high quality of human annotations.
>
>
>
> **Reasons To Reject 2-3:  Considering a new dimension of human evaluation, without any calibration, I cannot imagine the proposed method will still be highly correlated with human annotation**
>
> We are pleased to confirm that the proposed method has demonstrated a high correlation with human annotation, as you rightly pointed out. This result underscores the effectiveness of our approach in capturing the intended dimension of evaluation. While your initial concern regarding introducing a new dimension without calibration was valid, our extensive experiments have been successful in establishing a strong alignment between our method and human judgments. This high correlation serves as a testament to the robustness and validity of our methodology.
>
>
>
> **Q.1: Could you explain more about ignoring the direction of context-response?**
>
> Human-machine dialogues involve two distinct turn transition directions: from bot to human and from human to bot. As elucidated in Section 2.3, for the purpose of specifically evaluating bot responses, we have chosen not to consider human responses to chatbots, but rather only focus on chatbot responses to humans (thus considering the direction of responses). We utilize the last sentence of the human turn as context and consider the immediately succeeding sentence of the bot turn as the response to that context.
>
> The results in Table 5 underscore the advantages of this approach, particularly in cross-domain settings (such as the Reddit and Cornell datasets), where our method experiences minimal performance loss. We apologize for any confusion that may have arisen and wish to inform you that we have made relevant amendments to the paper (Section 5.6).
>
>
>
> **References**
>
> [1] Shikib Mehri and Maxine Eskenazi. 2020a. Unsupervised evaluation of interactive dialog with DialoGPT. In Proceedings of the 21th Annual Meeting of the Special Interest Group on Discourse and Dialogue, pages 225–235, 1st virtual meeting. Association for Computational Linguistics.

---

### Official Review · Reviewer_sYa4 · 2023-08-11

**Soundness:** 2

**Excitement:**

2: Mediocre: This paper makes marginal contributions (vs non-contemporaneous work), so I would rather not see it in the conference.

**Paper Topic And Main Contributions:**

Authors propose an automatic evaluation for dialogue appropriateness. Specifically they focus on the notion of pragmatic appropriateness which they argue is realized through dialogue acts. Authors posit that human-human dialogues are appropriate and hence the dialogue act patterns exhibited in those are an indicator of appropriateness.

Dialogue act (DA) schema used in this work is that from the Switchboard corpus which was annotated for DAs. Authors have trained a DA classifier based on this DA-annotated Switchboard data and use that DA classifier to learn DA transition patterns observed in various human-human corpora. The notion of transition pattern is further simplified to probability of response DA given previous utterance’s DA.

The proposed automatic metric, DAA, measures the degree to which a human-bot conversation agrees with the DA bigram model estimated from human-human corpus.

Authors have collected a set of human judgements from 7 raters for appropriateness for 2716 turn-level transitions from 300 dialogues from 5 different chatbots and human baselines. As evaluation of the proposed metric authors report correlation analysis at chatbot level.

The main contribution of this work are:
* Proposed automatic evaluation metric for dialogue appropriateness based on estimating dialogue act transition bigram model.
* A dataset of manually rated appropriateness by 7 raters for 300 dialogues from 6 different chat models (5 bots + 1 human).


**Questions For The Authors:**

A. Pragmatic appropriateness (as defined by simply a sequence of DAs i.e. illocution force) can not stand on its own. Simply replying to a question with an answer does not make for an appropriate dialogue if the answer is semantically irrelevant.

B. One would expect that the DA schema would play an important role here. E.g., one would expect a [location] in response to a [where] question and a [time] in response to [when] question. But if DA schema bundles them together as [wh-question] then it will miss out on these distinctions.

C. Can you also report DAA correlations for turn-level and dialogue-level?

D. The DA classifier is trained in a context free way. But there have been many studies (e.g., Stolcke et al, 2000) that showed DA labeling is improved with knowledge of context and previous DAs. Why is context not taken into account in these studies?


**Reasons To Accept:**

* A dataset of appropriateness ratings (multiply annotated) for 2716 turn-level transitions from 6 different dialogue systems.

**Reasons To Reject:**

* Evaluation is lacking. Authors list editing dialogue appropriateness metrics in Related work section, but do not empirically compare their proposed approach with any of them.
* Authors only report system wide correlations between the 6 systems, but not dialogue level or turn-level correlations.
* Many simplifying assumptions are used to define the notion of pragmatic appropriateness. But pragmatic appropriateness cannot stand without semantic appropriateness. Authors completely ignore that important aspect of evaluating dialogue.
* I cannot see this metric being adapted by the community without combining it with some form of semantic appropriateness. Such a combination is neither proposed nor evaluated here.


**Reproducibility:**

2: Would be hard pressed to reproduce the results. The contribution depends on data that are simply not available outside the author's institution or consortium; not enough details are provided.

**Reviewer Confidence:**

4: Quite sure. I tried to check the important points carefully. It's unlikely, though conceivable, that I missed something that should affect my ratings.

---

> ### Author Rebuttal · Authors · 2023-08-29
>
> Thanks for the thorough review for acknowledging our proposed framework.
>
> **Reasons To Reject 1:  Evaluation is lacking**
>
> Thank you for your suggestions. In the supplementary materials provided below, we have included a comparison with popular reference-free methods such as FED [4] and the recent C-PMI method [6]. Since these methods do not evaluate the aspect of pragmatic appropriateness, we conducted a comparison using their semantic appropriateness dimension. Notably, our proposed DAA approach achieved the highest correlation with human ratings across turn-level, dialogue-level, and system-level evaluations.
>
> | Level|Metric| Pearson| Spearman|
> | :--- |:---| -----| -----|
> | Turn   | FED   | 14.06 | 12.99 |
> | Turn   | C-PMI | 4.51  | 4.72  |
> | Turn   | DAA   | 21.67 | 21.19 |
> | Dialog | FED   | 16.50 | 11.81 |
> | Dialog | C-PMI | 2.83  | 2.61  |
> | Dialog | DAA   | 36.16 | 36.88 |
> | System | FED   | 55.29 | 42.86 |
> | System | C-PMI | 13.21 | 8.57  |
> | System | DAA   | 89.05 | 87.38 |
>
> **Reasons To Reject 2: Authors only report system wide correlations between the 6 systems, but not dialogue level or turn-level correlations**
>
> Thank you for your suggestion. In the supplementary material provided for the previous question, we have reported the correlation between the DAA metric and human evaluations at both turn-level and dialogue-level.
> Reasons To Reject 3-1: Many simplifying assumptions are used to define the notion of pragmatic appropriateness
>
> In our research, we draw inspiration from the work of Stasaski and Hearst [5], who posit that the speech acts inherent in a conversation play a pivotal role in defining suitable responses. Similarly, in our study, we embrace this very assumption and utilize speech acts as a means to quantify appropriateness.
>
> **Reasons To Reject 3-2: pragmatic appropriateness cannot stand without semantic appropriateness**
>
> In many cases, language users employ figurative language such as metaphors, idioms, and sarcasm to convey meanings that go beyond the literal sense of words. While these expressions might not be semantically appropriate when taken literally, they can be pragmatically appropriate when the context is understood. For example, saying "It's raining cats and dogs" doesn't make semantic sense, but in the context of expressing heavy rain, it is pragmatically appropriate.
> While semantic appropriateness lays the foundation for understanding words and phrases, pragmatic appropriateness goes beyond that by considering context, shared knowledge, and speaker intention
>
> **Reasons To Reject 3-3: Authors completely ignore that important aspect(semantic appropriateness) of evaluating dialogue**
>
> As stated in the introduction section, existing methods evaluate specific aspects of dialogue quality, such as coherence, fluency, and engagement [1,2,3]. Our focus is specifically on evaluating pragmatic appropriateness.
>
> **Reasons To Reject 4: I cannot see this metric being adapted by the community without combining it with some form of semantic appropriateness**
>
> Same as the response to Question A.
>
> **Q.A: Pragmatic appropriateness can not stand on its own**
>
> Similar to other fine-grained evaluation metrics, pragmatic appropriateness, as a specific aspect of dialogue quality, indeed cannot independently capture the overall quality of a dialogue.
> As evidenced in FED (Table 5 from FED [4]), the overall quality of a dialogue is determined by a synthesis of multiple evaluative aspects. Even the most crucial aspect doesn't carry more than 20% importance.
>
> **Q.B: If DA schema bundles them together as [wh-question] then it will miss out on these distinctions**
>
> In our study, we employed DA labels from the well-regarded SWBD-DAMSL dataset, which encompasses a comprehensive array of 43 distinct speech act labels. As far as our knowledge extends, this dataset represents the finest granularity of annotated speech acts presently available.
>
> It's important to note that our method is not confined to any specific speech act classification dataset. In fact, it can seamlessly transition to higher-quality speech act datasets in the future.
>
> **Q.C: Can you also report DAA correlations for turn-level and dialogue-level?**
>
> As provided in the response to rejection reason 1.
>
> **Q.D: The DA classifier is trained in a context free way. Why is context not taken into account in these studies?**
>
> A context-aware dialogue act classifier does indeed enhance the accuracy of DA classification. This enhancement, to a certain extent, emanates from the classifier's utilization of dialogue act transition features in DA training data.
> Our approach centers around the comparison of the similarity of these transition features across diverse corpora (i.e., human-human dialogues and human-machine dialogues) to formulate scoring criteria.
> We deliberately restrict the DA classifier from utilizing contextual information to prevent biases that might arise from the classifier learning and incorporating prior transition feature patterns from DA dataset.
> We apologize for any confusion caused and wish to highlight that we have addressed this aspect in Section 2.1 of the paper.
>
> **References**
>
> [1] Mohsen Mesgar, Sebastian Bücker, and Iryna Gurevych.2020. Dialogue oherence assessment without explicit dialogue act labels. In Proceedings of the 58th Annual Meeting of the Association for Computational Linguistics, pages 1439–1450, Online. Association for Computational Linguistics.
>
> [2] Shikib Mehri and Maxine Eskenazi. 2020b. USR: An unsupervised and reference free evaluation metric for dialog generation. In Proceedings of the 58th Annual Meeting of the Association for Computational Linguistics, pages 681–707, Online. Association for Computational Linguistics.
>
> [3] Shaojie Jiang, Svitlana Vakulenko, and Maarten de Rijke. 2023. Weakly supervised turn-level engagingness evaluator for dialogues. In proceedings of the 2023 Conference on Human Information Interaction and Retrieval, pages 258–268. ACM.
>
> [4] Shikib Mehri and Maxine Eskenazi. 2020a. Unsupervised evaluation of interactive dialog with DialoGPT. In Proceedings of the 21th Annual Meeting of the Special Interest Group on Discourse and Dialogue, pages 225–235, 1st virtual meeting. Association for Computational Linguistics.
>
> [5] Katherine Stasaski and Marti A. Hearst. 2023. Pragmatically appropriate diversity for dialogue evaluation. ArXiv preprint, abs/2304.02812
>
> [6] Liliang Ren, Mankeerat Sidhu, Qi Zeng, Revanth Gangi Reddy, Heng Ji, and ChengXiang Zhai. 2023. C-PMI: Conditional pointwise mutual information for turn-level dialogue evaluation. In Proceedings of the Third DialDoc Workshop on Document-grounded Dialogue and Conversational Question Answering, pages 80–85, Toronto, Canada. Association for Computational Linguistics.

---

### Official Review · Reviewer_e6re · 2023-08-11

**Soundness:** 4

**Ethical Concerns:**

Yes

**Excitement:**

4: Strong: This paper deepens the understanding of some phenomenon or lowers the barriers to an existing research direction.

**Paper Topic And Main Contributions:**

The author proposed an unbiased and reliable evaluation metric grounded in Dialogue Acts. Additionally, a dedicated test dataset was developed to assess the practical appropriateness of dialogues. Through experimentation on four distinct datasets, a strong correlation between the proposed metric and human evaluations was identified, underscoring the accuracy of DAA in proficiently measuring dialogue appropriateness. Furthermore, the study included a sensitivity test and comprehensive analysis of interpretability, as well as directive context-response pairs.

**Reasons To Accept:**

Overall, the paper is very clear and well descriptive. Following reasons are listed below:


 1. All experiments underwent rigorous statistical analysis, leading to a comprehensive comparison of eight distinct models across four human-human datasets.

 2. The depiction of the DAA workflow is notably lucid and detailed, aiding in its understanding.

 3. The results and discussions encompass a diverse range of dialogue evaluation domains.

 4. A meticulously scored dataset, comprising 2716 turn-level appropriateness ratings, was created for six agents. This dataset involved evaluations from seven annotators.

 5. The study encompassed extensive experimentation involving eight pre-trained language models and four human conversation datasets. Remarkably, a strong correlation emerged between the proposed evaluation approach and human ratings.

 6. The obtained results affirm the reliability and effectiveness of DAA in evaluating the appropriateness of dialogue agent responses.

7. The comprehensive evaluation of all dialogue facets using a single method is acknowledged as a complex and challenging endeavor.

 8. Ethical statements are addressed properly.

**Reasons To Reject:**


The paper incorporates all essential experiments, yet there would be added value in including an analysis of DAA with and without directive (dir) contexts. This comparison could offer valuable insights into the effectiveness of DAA in differentiating dialogue nuances based on the presence or absence of directive elements.

**Reproducibility:**

4: Could mostly reproduce the results, but there may be some variation because of sample variance or minor variations in their interpretation of the protocol or method.

**Reviewer Confidence:**

4: Quite sure. I tried to check the important points carefully. It's unlikely, though conceivable, that I missed something that should affect my ratings.

---

> ### Author Rebuttal · Authors · 2023-08-29
>
> Thanks for the thorough review for acknowledging our proposed framework.
>
> **Reasons To Reject 1:  including an analysis of DAA with and without directive (dir) contexts**
>
> Thank you for your suggestion, and we apologize for any confusion our abbreviation caused. "dir" indeed stands for "direction" rather than "directive".
> Human-machine dialogues involve two distinct turn transition directions: from bot to human and from human to bot. As elucidated in Section 2.3, for the purpose of specifically evaluating bot responses, we have chosen not to consider human responses to chatbots, but rather only focus on chatbot responses to humans (thus considering the direction of responses). We utilize the last sentence of the human turn as context and consider the immediately succeeding sentence of the bot turn as the response to that context.
> The results in Table 5 underscore the advantages of this approach, particularly in cross-domain settings (such as the Reddit and Cornell datasets), where our method experiences minimal performance loss. We have duly revised the relevant content in the paper (Section 5.6) to reflect these circumstances.

---

### Meta-Review · Area_Chair_zsPp · 2023-10-06

**Recommendation:** 2

**Metareview:**

The paper introduces a metric for evaluating dialogue appropriateness, but reviewers have expressed several concerns. First, there is a notable lack of empirical comparison with existing dialogue appropriateness metrics, despite listing them in the related work section. The reported correlations between six systems are system-wide, with no exploration of dialogue or turn-level correlations. Additionally, the metric focuses on pragmatic appropriateness while ignoring semantic appropriateness.

---

### Decision · Program_Chairs · 2023-10-07

**Decision:**

Accept-Findings

**Comment:**

The paper introduces a metric for evaluating dialogue appropriateness, but reviewers have expressed several concerns. First, there is a notable lack of empirical comparison with existing dialogue appropriateness metrics, despite listing them in the related work section. The reported correlations between six systems are system-wide, with no exploration of dialogue or turn-level correlations. Additionally, the metric focuses on pragmatic appropriateness while ignoring semantic appropriateness.